# Genome-wide sequencing-based identification of methylation quantitative trait loci and their role in schizophrenia risk

Kira A. Perzel Mandell [1,2], Nicholas J. Eagles [1], Richard Wilton[3], Amanda J. Price [1,2], Stephen A. Semick [1], Leonardo Collado-Torres [1], William S. Ulrich[1], Ran Tao[1], Shizhong Han[1,4], Alexander S. Szalay[3,5], Thomas M. Hyde [1,4,6], Joel E. Kleinman [1,4], Daniel R. Weinberger [1,2,4,6,7 ✉] & Andrew E. Jaffe [1,2,4,7,8,9 ✉]

DNA methylation (DNAm) is an epigenetic regulator of gene expression and a hallmark of gene-environment interaction. Using whole-genome bisulfite sequencing, we have surveyed DNAm in 344 samples of human postmortem brain tissue from neurotypical subjects and individuals with schizophrenia. We identify genetic influence on local methylation levels throughout the genome, both at CpG sites and CpH sites, with 86% of SNPs and 55% of CpGs being part of methylation quantitative trait loci (meQTLs). These associations can further be clustered into regions that are differentially methylated by a given SNP, high-lighting the genes and regions with which these loci are epigenetically associated. These findings can be used to better characterize schizophrenia GWAS-identified variants as epi-genetic risk variants. Regions differentially methylated by schizophrenia risk-SNPs explain much of the heritability associated with risk loci, despite covering only a fraction of the genomic space. We provide a comprehensive, single base resolution view of association between genetic variation and genomic methylation, and implicate schizophrenia GWAS-associated variants as influencing the epigenetic plasticity of the brain.

[1] Lieber Institute for Brain Development, Johns Hopkins Medical Campus, Baltimore, MD, USA. [2] Department of Genetic Medicine, Johns Hopkins University School of Medicine (JHSOM), Baltimore, MD, USA. [3] Department of Physics and Astronomy, Johns Hopkins University, Baltimore, MD, USA. [4] Department of Psychiatry and Behavioral Sciences, JHSOM, Baltimore, MD, USA. [5] Department of Computer Science, JHSOM, Baltimore, MD, USA. [6] Department of Neurology, JHSOM, Baltimore, MD, USA. [7] Department of Neuroscience, JHSOM, Baltimore, MD, USA. [8] Department of Mental Health, Johns Hopkins Bloomberg School of Public Health (JHBSPH), Baltimore, MD, USA. [9] Department of Biostatistics, JHBSPH, Baltimore, MD, USA. ✉email: drweinberger@libd.org; andrew.jaffe@libd.org

DNA methylation (DNAm) plays an important role in the epigenetic regulation of gene expression. It varies throughout development and among tissue types, and has been thought to be a high-fidelity representation of the interaction between genes and environment. While some variation in DNAm can be attributed to developmental and exogenous factors, such as diet[1] and cigarette smoking[2], Davies et al.[3] identified some inter-individual variation that is consistent across tissue types. This provided evidence that DNA sequence drives DNA methylation levels, at sites known as methylation quantitative trait loci (meQTLs). Inter-individual DNAm differences have since been confirmed by twin studies[4,5]. Initial studies found methylation association with sequence variants at specific loci[6]. These epigenetic associations likely extend beyond losing the "C" or "G" allele in CpG dinucleotides (for example, through deamination of the cytosine base in this genomic context)[7].

Genome-wide studies are necessary to fully understand the extent and genomic properties of meQTLs. However, so far, most large-scale studies have used microarray technologies that only measure a small proportion of CpGs[8–10]. The largest study to date to test associations between genotype and DNAm used MBD-seq, a method lacking single base-pair resolution[11]. Yet even with limited resolution, these initial studies have found that local genetic influence on DNAm is extensive throughout the genome, and meQTLs are enriched at regulatory sites[12,13].

Currently, a major puzzle in the field of functional genomics is understanding the molecular effects of genetic risk loci and variants identified by genome-wide association studies (GWAS) for many common disorders and traits which do not involve coding sequences. This is particularly challenging in tissues like brain that are difficult to access or model, leaving little clarity into genetic mechanisms behind psychiatric disorders such as schizophrenia (SCZD). While schizophrenia is highly heritable[14], and GWAS have identified a growing number of significant loci[15,16], only few loci have been functionally resolved[17]. Genome-wide gene expression QTL (eQTL) approaches[18,19], and their extensions[20–22], have prioritized variants and associated genes, but many genomic loci fail to associate with nearby gene expression. In contrast, associating schizophrenia risk variants with a stable epigenetic mark like DNAm provide clues for potential epigenetic mechanisms of action[23,24]. Indeed, previous meQTL maps using microarray technologies implicated a larger number of SCZD risk loci than eQTL maps, while only measuring a fraction of the methylome[10]. DNAm itself may further reflect the cumulative effects of environmental exposures across the lifespan[25], and may represent a surrogate of "E" in GxE interactions that contribute to risks for many disorders[26] that can further act as a mediator of genetic risk on gene expression.

Unlike microarray technologies, whole-genome bisulfite sequencing (WGBS) has the advantage of measuring cytosine methylation at single base-pair resolution, as well as measuring CpH (H = A, T, or C) DNA methylation levels (in addition to CpGs). While CpH sites are generally unmethylated in somatic tissues, neurons in the human brain have uniquely high levels of CpHm[27]. By leveraging this technology, we have created the most extensive genomic meQTL map in human postmortem brain tissue to date, and use this information to fine-tune our understanding of the molecular mechanisms of genetic and epigenetic risk for schizophrenia.

## Results

### Components of global variation in large-scale WGBS data sets.
We performed whole-genome bisulfite sequencing (WGBS) to gain a comprehensive view of genetic influence on DNAm in the adult human brain using two brain regions: the hippocampus and

the dorsolateral prefrontal cortex (DLPFC). These regions have been prominently implicated in the pathogenesis of many psychiatric disorders, particularly schizophrenia[28]. After data processing and quality control (see "Methods" section), we analyzed 165 DLPFC samples and 179 hippocampal samples from a total of 183 adult donors aged 18–96 years (161 donors had data from both regions, Supplementary Dataset 1). Data were generated across two large diagnosis- and region-balanced batches. We assessed 29,401,795 CpG sites across the epigenome, with an average post-processing coverage of 17.3 reads per CpG site. 78% of sites were highly (>80%) methylated while a minority were lowly or unmethylated (8% are <20%, Supplementary Fig. S1). While the technical effects of measuring DNAm levels using microarrays are well-established[29], particularly in human brain tissue[10], corresponding assessments using WGBS data have been limited due to available comparisons being relatively small studies. We, therefore, assessed the contributions of different biological and technical variables on genome-wide CpG DNAm levels measured with WGBS.

First, we performed principal component analysis (PCA) across the raw DNAm levels of the million most variable CpGs. The top principal components were associated with quantitative/genotype-defined ancestry (PC1: 6.5% variance explained, Supplementary Fig. S2), estimated neuronal fraction (PC2: 3.34%), processing batch (PC4: 1.37%), and brain region (PC5: 0.84%). The major batch effects resulted from the inclusion of ENCODE "blacklist" regions[30], which have been reported to cause problems with mapping and alignment in high-throughput sequencing data, particularly epigenomic data. These processing issues are likely further exacerbated in WGBS data, where the bisulfite treatment results in lower complexity libraries depleted of cytosines, which presumably relates to the influence of blacklist regions and ancestry on DNAm levels. Cytosines in these black-listed regions were therefore removed from reported site-specific analysis results. Another increasingly common step in WGBS data processing involves "smoothing" local CpG DNAm levels within each sample to improve precision and borrow strength across nearby CpGs[31]. Smoothing reordered the top components of variation (Supplementary Fig. S3), and resulted in the top component of variation representing both batch and estimated neuronal fraction (both PC1 and PC2, explaining 13.9% and 10% of the variance, respectively). Previous analyses of Illumina microarray-derived adult homogenate DLPFC data suggested that estimated neuronal fraction was the largest component of (CpG) DNAm level variation[10]. Microarray technology implicitly produces somewhat smooth DNAm levels for a large fraction of probes that target multiple CpG sites. While the effects of the brain region were further magnified with smoothing, the effects of quantitative ancestry were dampened, and became associated with PC4 (1% variance explained) rather than PC1 of raw DNAm levels (6% explained variance; Fig. 1a–e).

We further complemented these global analyses using site-specific variance components analysis, estimating the percentage of smoothed DNAm level variance explained by technical and biological components at each autosomal CpG, excluding the blacklist ($N = 26,416,185$, using ANOVA, see "Methods" section, Fig. 1f, see "Data availability" section). The factors that explained the largest components of site-specific variation were technical batch (median: 9.6% variance explained, interquartile range 3.4–19%), and related flow cell (7.9%, 6.3–9.6%) and instrument (2.9%, 2.1–3.2%) variables, as well as the more biological brain region (1.5%, 0.3–5.9%), estimated neuronal fraction (1%, 0.2–3.8%) variables. Traditionally considered confounders in postmortem human brain studies—including tissue pH and postmortem interval (PMI)—had very little influence on site-specific DNAm levels using WGBS (in line with previous microarray-based analyses[10]). For example, pH and PMI

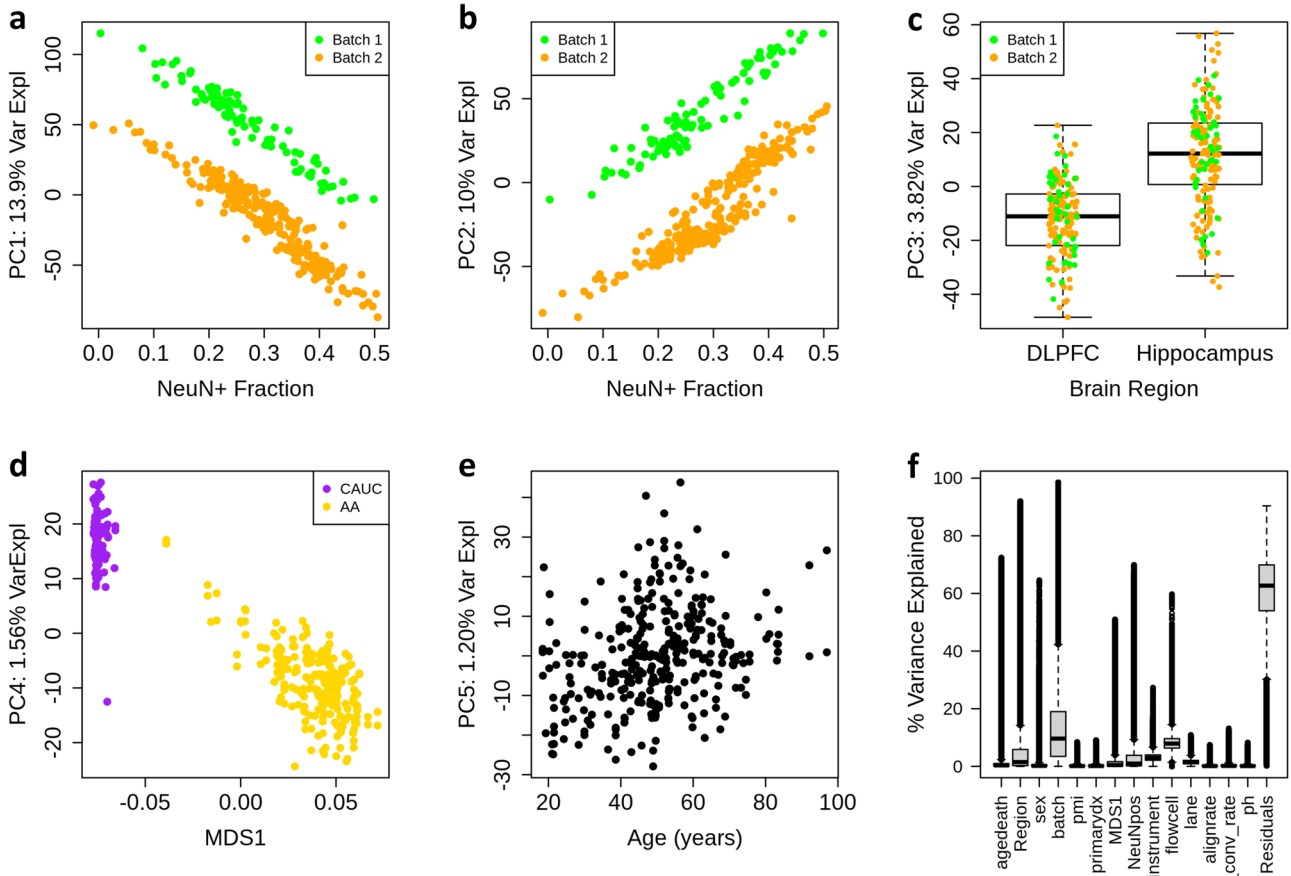

**Fig. 1 Variance in smoothed methylation data, post-QC.** PCA was performed on all sites excluding the sex chromosomes and ENCODE's blacklist. **a, b** We find that the top principal components of smoothed methylation data associate with both batch and neuronal composition. **c** We see that the third principal component is associated with the brain region, and no longer associated with the batch. **d** In smoothed methylation data, ethnicity is reduced to the 4th principal component. MDS: multidimensional scaling of genotype data. **e** Age associates with the fifth principal component. **f** Variance explained was analyzed using ANOVA by each individual CpG site. We see that brain region and batch effects explain a large deal of variance, while biological factors such as PMI and pH explain very little. $n = 26,416,185$ CpGs.

explained more than 1% of the variance across only approximately 5% of measured sites. Other technical variables hypothesized to influence DNAm levels like the sequencing alignment rates and bisulfite conversion rates (estimated with λ spike-in sequences, see "Methods" section) showed little influence in this analysis. Overall, there was an extensive residual variation of DNAm levels for the majority of sites beyond these technical and biological variables.

**Local genetic variation has strong effects on CpG DNA methylation levels**. We hypothesized that a large component of this residual DNAm variation was likely captured by local genetic sequence. We, therefore, performed genome-wide methylation quantitative trait locus (meQTL) analyses (see "Methods" section) on smoothed DNA methylation levels in each brain region separately (across 29,401,795 CpG sites). In the DLPFC, we computed meQTLs between each of these CpGs and the subset of common SNPs within 20 kb upstream and downstream, which identified 482,579,961 significant SNP–CpG pairs (at FDR < 0.01, see "Data availability" section), representing 6,807,821 (86%) of the tested SNPs and 14,551,080 (55%) of tested CpGs. Sensitivity analyses using the more stringent Bonferroni cutoff (corresponding to ~3 billion tests)—assuming both CpGs and SNPs are independent (which is likely an overly-stringent assumption given known spatial autocorrelation of CpGs and linkage disequilibrium of SNPs)—identified 101,482,392 SNP–CpG pairs,

representing 15% of CpGs and 37% of tested SNPs. Given the high genomic correlation among both CpGs and SNPs, we performed the same analysis with a set of 535,859 LD-independent SNPs ($R^2 < 0.2$) to reduce the potential effects of linkage disequilibrium (LD) potentially inflating these statistics. This sensitivity analysis found a substantial proportion of CpGs (8,390,092, 29%) and SNPs (402,407, 75%) identified as meQTLs at FDR < 0.01 with similar properties. Most SNPs associated with methylation levels at many nearby CpG sites (mean = 57 CpGs, median = 43 CpGs), and the methylation-associated SNPs had varying genomic widths of effect in this local window ranging from 1 bp to the full 40 kb (mean = 14.5 kb, median = 12.7 kb). Effect sizes were generally small, with a mean of 2.6% change in methylation level per allele (IQR: 1.4–3.1%), but ranging up to 47%. Enforcing more stringent Bonferroni significance among meQTLs resulted in higher effect sizes (mean = 4.5%), lower widths of effect (mean = 7.5 kb), and fewer—though still numerous—CpGs associated with each SNP (mean = 34). In both analyses we found that SNPs that disrupt a CpG dinucleotide (i.e., a variant at the C or G, which would ablate the capacity for methylation) have a slightly but significantly lower width of effect and a slightly higher number of correlated CpGs, meaning they have a higher effect density. This may be attributed to the fact that CpGs tend to cluster in the genome. Additionally, despite tested SNPs being LD-independent in the second analysis, we find that half of CpGs associate with more than one SNP, with

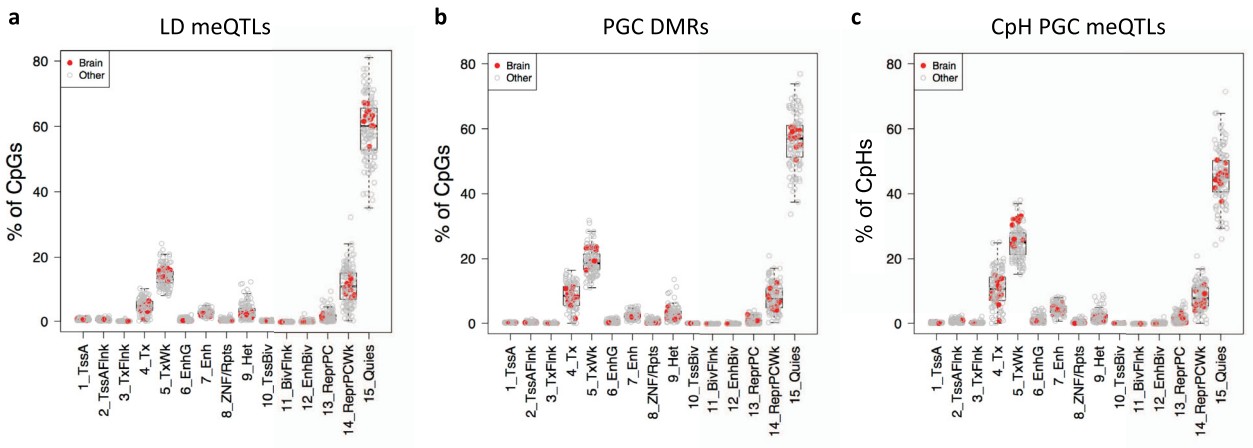

**Fig. 2 Chromatin state of meQTLs and gDMRs.** We assessed the chromatin state of sites identified as meQTLs and gDMRs using data from various tissues from the Roadmap Epigenomics Project. Data for brain tissues are highlighted in red. $n = 127$ tissue types. Chromatin states are defined as follows. 1: Active TSS, 2: Flanking Active TSS, 3: Transcription at gene 5′ and 3′, 4: Strong transcription, 5: Weak transcription, 6: Genic enhancers, 7: Enhancers, 8: ZNF genes & repeats, 9: Heterochromatin, 10: Bivalent/Poised TSS, 11: Flanking Bivalent TSS/Enhancer, 12: Bivalent Enhancer, 13: Repressed PolyComb, 14: Weak Repressed PolyComb, 15: Quiescent/Low. **a** In genome-wide meQTLs, assessing a set of LD-independent SNPs, we see that the vast majority of meQTL-CpGs are in quiescent chromatin regions. **b** We see that compared to genomic SNPs, CpGs associated with PGC schizophrenia risk SNPs are enriched for regions of active transcription and depleted for regions of quiescent chromatin. **c** We see that CpH sites associated with PGC schizophrenia risk SNPs are often in regions of active transcription.

a mean and median of 2, in line with previous observations in gene expression data[18,32]. Using information from the Roadmap Epigenome[33], most genetically-associated CpGs were in quiescent genomic regions, and depleted for enhancer regions in the human brain (Fig. 2a). The meQTLs from both brain regions are searchable by SNPs or genomic regions of interest at: https://eqtl.brainseq.org/WGBS_meQTL/.

Analogous results were observed in hippocampus samples, yielding 505,142,175 significant pairs that represented 14,647,533 CpGs (55%) and 6,900,009 SNPs (87%, see "Data availability" section). Analyses on LD-independent SNPs yielded a near-similar proportion of SNPs (403,373 SNPs, 77%) implicating a similar number of CpGs as seen in the DLPFC (8,566,898). Effect sizes were similarly small, with a mean of 2.6% change in methylation level per allele. Hippocampal meQTLs have similar width of effect as those in DLPFC with a mean of 15,661 bp and an average of 60 CpGs associated with a SNP. These analyses suggested that the global properties of meQTLs were highly similar across brain regions. Given the difficulty in identifying suitable/comparable brain region-specific replication data sets, we treated one region as the discovery data set and the other as replication, and calculated $\pi_1$ statistics. By comparing our findings between the two brain regions, we found very high replicability, with hippocampal meQTLs showing a very high sharing ($\pi_1 = 0.978$) with significant meQTLs identified in DLPFC (using chr1, see "Methods" section).

We next performed a series of secondary analyses to better characterize the determinants of such extensive genetic regulation of DNA methylation. First, due to the mixed ethnicities of our samples, and the potentially large differences in allele frequencies between ancestry groups[34], we ran post-hoc meQTL analysis on a subset of meQTLs identified in full genome analysis in the DLPFC, separating samples into two groups by self-reported race. African Americans (AAs) made up 67% of our total sample, and thus were more likely to drive the results. Using significant meQTLs on chromosome 1, analyses using only African American samples ($N = 112$) showed that 99.91% of the meQTLs were directionally consistent with the full analysis, with 92% marginally significant ($P < 0.05$) and 66% genome-wide significant (FDR < 0.01) in the smaller sample size and an overall

sharing of $\pi_1 = 0.995$. In the European ancestry samples ($N = 53$), of the 90% of meQTLs that had polymorphic SNPs in this group, 95% of meQTLs were directionally consistent, with 55% marginally significant and 16% genome-wide significant with an overall sharing of $\pi_1 = 0.801$. These decreased proportions compared to AA-specific analyses at least partially related to the smaller sample size and resulting in decreased statistical power (Supplementary Fig. S4). We also found that in general, differences in minor allele frequencies across ancestry groups did not associate with differences in meQTL effect magnitude (Supplementary Fig. S5), indicating that differences in ethnicity composition of our samples were likely not driving our combined ancestry analyses above. As an additional sensitivity analysis, we performed meQTL analysis within each diagnostic group separately to assess the extent to which schizophrenia diagnosis influenced our findings. We found very high sharing ($\pi_1 > 99\%$) among significant meQTLs discovered in either diagnostic groups (and assessed in the other/non-discovery group), supporting that meQTL effects were globally similar between diagnostic groups. We further explored the robustness of the selected meQTL window size (20 kb) using heritability analysis (see "Methods" section) on the methylome[35] with different window sizes (20, 100, 500 kb). DNA methylation levels were highly heritable using a 20 kb window size, with 38% of tested CpG sites showing significant heritability (FDR < 0.01). These heritability results were further consistent with the above meQTL analyses, with 99% of significantly heritable CpGs being meQTLs (and conversely, 68% of meQTL-CpGs were heritable). Larger window sizes in heritability analysis actually identified fewer CpGs with significantly heritable methylation, implying that most genetic control of methylation acts in *cis* and confirming that our meQTL testing window was comprehensive. These WGBS data further replicated meQTLs identified in Illumina 450k microarray data ($\pi_1 = 0.67$)[10], even though CpGs profiled on this microarray platform were depleted for meQTLs (~30% of CpGs) compared to WGBS (~55% of CpGs), in line with gene-biased designs of Illumina microarrays. We, therefore, identified widespread genetic control of CpG methylation levels. Understanding the details of this landscape may help elucidate the functional significance of SNPs highlighted by GWAS.

**Widespread meQTLs among schizophrenia risk variants**. DNA methylation previously has been shown to play a role in mediating genetic risk for neuropsychiatric (and other common) disorders[10,36–38], but all previous meQTL analyses have utilized microarray, not sequencing-based, methylome data. We performed extensive meQTL analyses on SNPs associated with genetic risk for schizophrenia in these large WGBS data sets. We specifically performed chromosome-scale meQTL analysis using each of the "index" SNPs for loci associated with schizophrenia from the most recent GWAS study of schizophrenia, i.e., PGC2 + CLOZUK[15]. We assessed index SNPs with high-quality genotype data in each region - 152 SNPs in DLPFC and 153 in the hippocampus. Each SNP was tested against every CpG site in the genome, considering a distance of <250 kb *cis* and everything else *trans*. In DLPFC we found 25,382 significant (FDR < 0.01, Supplementary Dataset 2) SNP–CpG pairs, representing 147 SNPs and 25,303 CpGs, showing that most Psychiatric Genomics Consortium (PGC) loci contain SNPs that associate with local DNA methylation levels (as only 107 SNP–CpG pairs were in *trans*). Schizophrenia risk-associated SNPs on average associated with 172 CpGs (median = 104), and in this *cis* window had an average genomic width of the effect of 177 kb (median = 147 kb).

We further performed functional validation of these associations using corresponding gene expression data. Using RNA-seq data from the same regions and donors (see "Methods" section), we assessed whether methylation at these CpGs correlated with neighboring expression levels. Using previous eQTL analyses on these same PGC loci[18,39], we assessed the mediation of expression by mCpG (see "Methods" section). Eleven of 127 loci had a correlation between gene expression and the methylation with which they are associated. Importantly, 10 of these associated with at least one CpG that mediated expression by at least 25%. The same analyses on the exon and junction levels picked up subtler effects, detecting 18 and 27 loci mediating expression levels via methylation, respectively. We found that overall, methylation mediation was most potent on the exon level (median = 40%), then the junction level (median = 32%), and least potent on the full gene level (median = 23%), in line with the putative role of DNAm in promoting gene splicing[40].

The same meQTL analysis was performed in the hippocampus WGBS data, revealing 48,023 significant SNP–CpG pairs (Supplementary Dataset 3), representing 139/153 tested SNPs (including 15,119 *trans*-meQTLs, 31.5%). Within the subset of significant DLPFC meQTLs, hippocampal meQTLs had an overall sharing of $\pi_1 = 0.97$, indicating that our findings are very consistent between brain regions.

These results indicated that meQTL effects, at least in the context of GWAS associations with schizophrenia, have much broader effects than traditionally considered, and much wider than the 20 kb window examined at the full genome level. In order to see if schizophrenia-associated meQTLs are comparable to non-disease-associated meQTLs, we took 5000 random SNPs representing all levels of MAF and ran meQTL analysis with a 250 kb window. Again we found that the majority of SNPs (93%) are meQTLs. We also find that neither MAF nor population-MAF differences associate with any meQTL characteristics. Interestingly, we found that these random meQTLs had significantly lower width of effect than schizophrenia-associated meQTLs in both regions (DLPFC $P = 8.9e{-}5$, Hippocampus $P = 1.1e{-}11$), and a significantly fewer number of affected CpGs in the hippocampus ($P = 0.002$). This combined with the chromatin state enrichment analysis below may indicate that these PGC-meQTLs are particularly functional, and potentially involved in disease processes, as opposed to just being standardly representative of the whole genome.

**Risk-associated meQTL effects cluster in the genome**. We then proceeded to cluster our meQTL-CpGs into genetic differentially methylated regions (gDMRs)—regions where methylation is differential by additive genotype—for better functional characterization. Using a CpG-specific *t*-statistic cutoff of 5 (see "Methods" section), these sites could be clustered into 1277 gDMRs (Fig. 3 and Supplementary Dataset 4). The majority of SCZD index SNPs had such gDMRs, and most had more than one (mean = 9.5, median = 6). The overall span of effect for each SNP was much larger than the 20 kb *cis* window we tested above for meQTL analyses across the full genome, ranging up to 240 Mb on a single chromosome, with a median of 95 kb (mean = 17.5 Mb). Using Roadmap Epigenome[33] data, these SCZD risk-associated gDMRs were enriched over the background of genome-wide LD-independent meQTLs for transcriptional and weak transcriptional chromatin signatures (Fig. 2b). They were also comparatively depleted for weak repressive polycomb and quiescent chromatin signatures. Overall, these gDMRs were in or near genes enriched for GO terms related to synapse and membrane potential (Supplementary Fig. S6). 20 gDMRs overlapped with psychENCODE enhancers, and 142 overlapped with promoter regions. The genes connected to these promoters were enriched for GO terms related to acetylcholine, ion channels, and neurotransmitters.

Overall, the results of clustering meQTL-CpGs were quite similar between regions. In the hippocampus, there were 1408 gDMRs (Supplementary Dataset 5), and more than half (853) of gDMRs directly overlapped with a gDMR in the DLPFC. Furthermore, 95.1% of DLPFC-identified gDMRs showed marginal $P < 0.05$ significance among hippocampal meQTL effects (based on the average *P*-value for each gDMR). In general, allelic association with methylation in the hippocampus correlated with the association in DLPFC gDMRs ($r^2 = 0.69$, Supplementary Fig. S7). A strong majority of gDMRs were located inside introns. Again we found that most SNPs have multiple associated gDMRs (mean = 10, median = 7) and have a high total genomic width (mean = 15,447,206 bp, median = 122,887 bp). Only 16 of these gDMRs contain the actual GWAS index SNP, suggesting that these effects are more than just local consequences of genetic variations. Overall, the genes represented by these gDMRs have enrichment for GO terms related to synapses, membrane potential, and inositol triphosphate (IP3), a second messenger signaling molecule. In both regions, a handful of gDMRs had a correlation to the expression of some gene (see "Methods" section), and a few correlated to many genes, but most of these gene-gDMR pairs were on different chromosomes, making results difficult to interpret. Most pairs were negatively correlated though, which fits with the traditional understanding of the suppressive effect of methylation on gene expression[41,42].

These SCZD risk-associated gDMRs were further used as input to partitioned LDSC analysis (see "Methods" section) to illuminate their clinical relevance[43]. As a first pass, we ran partitioned LDSC analysis on the LD blocks of the schizophrenia GWAS loci used in meQTL analysis, which tested for enrichment of top loci versus the rest of the genome, controlling for CNS-relevant functional loci (see "Methods" section). These significant GWAS loci LD blocks explained 15% of the additive genetic heritability explained by SNPs ($h^2_{SNP}$), with a 10-fold enrichment over the full genome ($P = 8e{-}16$), in line with being the top-ranked loci in the GWAS. We then considered two sets of gDMRs defined by two different statistical thresholds: a more liberal *t*-statistic > 3.5 cutoff (which corresponded roughly to controlling an FDR < 0.01 in *cis* meQTL analysis), termed DMRs[3.5], and the subset of entirely-contained gDMRs defined by $t > 5$ (from above), termed DMRs[5]. The DMRs[3.5] were generally larger and more distant from each index SNP than the DMRs[5], with the

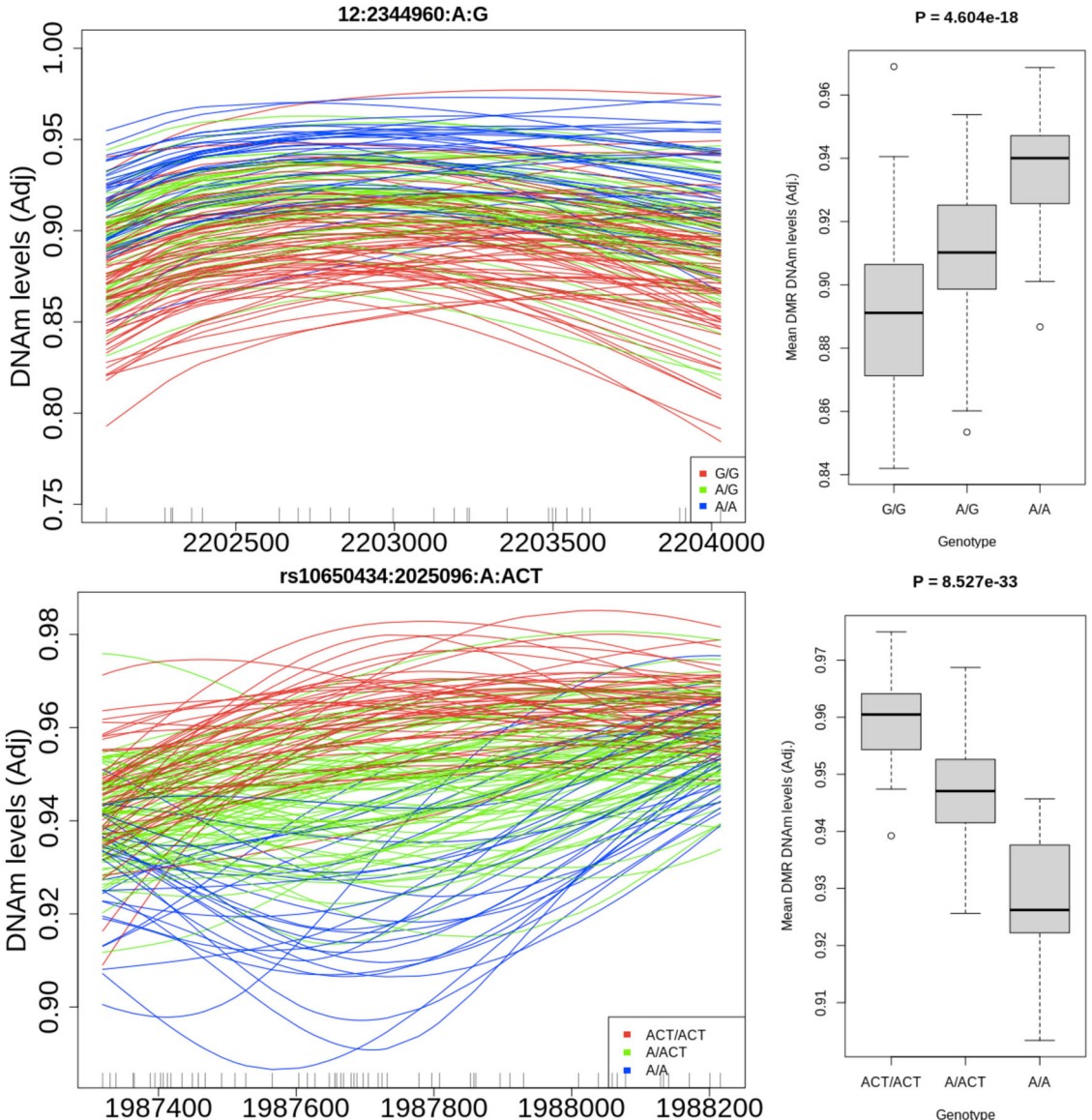

**Fig. 3 Schizophrenia risk-associated gDMRs.** Two examples of regions where methylation levels are associated with genotype at a schizophrenia risk associated SNP.

majority of DMRs[3.5] distal (in *trans*). We further divided these gDMR sets into those *cis* (DMRs5[cis] and DMRs3.5[cis]) and *trans* (DMRs5[trans] and DMRs3.5[trans]) relative to the PGC loci, i.e., those that were within the GWAS LD blocks, and those outside these blocks (Fig. 4). First, by comparing the heritability estimates from *cis* versus *trans* gDMRs at both cutoffs (i.e., DMRs5[cis] versus DMRs5[trans]), we found that the majority of schizophrenia heritability and enrichment was driven by *cis* regions. For example, among the DMRs3.5, the subset that was *cis* (DMRs3.5[cis]) explained 12% of $h^2_{SNP}$, with a 156-fold enrichment over the whole genome ($P = 1.3e-14$), and were further highly enriched compared to the background of the overall GWAS-significant LD blocks. Approximately 80% of all *cis* $h^2_{SNP}$ of the GWAS-significant loci (LD blocks) were captured by DMRs3.5[cis] (12% versus 15%), even though they contained only 3% of loci sequence (1.65 Mb versus 56.5 Mb). In contrast, DMRs3.5[trans] only explained 1.7% of $h^2_{SNP}$, and were not significantly enriched for schizophrenia risk (($P = 0.14$); Fig. 4). Despite representing a very small portion of the genome (658 kb), the more stringent *cis* gDMRs still explained 8.7% of $h^2_{SNP}$, with very strong enrichment

(243-fold, $P = 1e-10$). At only 48 kb, the stringent *trans* gDMRs were not wide enough to effectively detect enrichment, and only explained 0.5% of $h^2_{SNP}$. These results together suggest that the majority of schizophrenia genetic risk in these genome-wide significant loci specifically localizes among small subsets of genomic regions associated with proximal/nearby DNAm levels.

**Genetic regulation of CpH DNAm levels in homogenate brain tissue**. While non-CpG (CpH) DNA methylation predominantly occurs in neuronal cells in the human brain[27], we could nevertheless observe detectable levels in homogenate/bulk tissue (which contains 20–40% neuronal cells[44]). We analyzed 64,806,159 CpH sites in the DLPFC and 34,909,109 CpH sites in the hippocampus, after filtering for only sites which had at least moderate coverage and non-zero methylation levels across samples (see "Methods" section). These numbers of observable sites in homogenate tissue from adult donors were much larger than in prenatal donors, as CpH methylation occurs in post-mitotic neurons[45], and comparable to smaller studies of neuronal nuclei

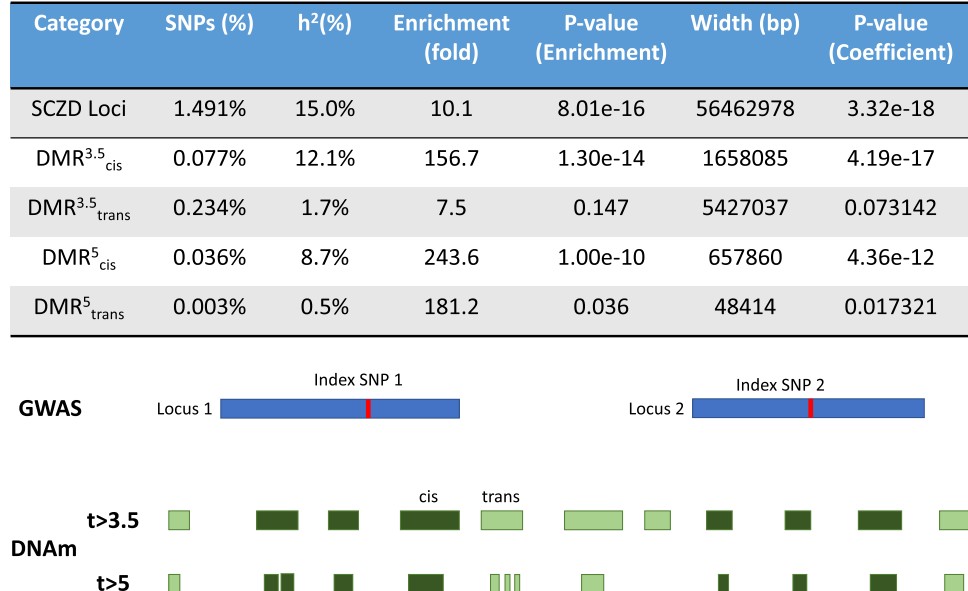

| Category | SNPs (%) | $h^2$(%) | Enrichment (fold) | P-value (Enrichment) | Width (bp) | P-value (Coefficient) |
|---|---|---|---|---|---|---|
| SCZD Loci | 1.491% | 15.0% | 10.1 | 8.01e-16 | 56462978 | 3.32e-18 |
| $DMR^{3.5}_{cis}$ | 0.077% | 12.1% | 156.7 | 1.30e-14 | 1658085 | 4.19e-17 |
| $DMR^{3.5}_{trans}$ | 0.234% | 1.7% | 7.5 | 0.147 | 5427037 | 0.073142 |
| $DMR^5_{cis}$ | 0.036% | 8.7% | 243.6 | 1.00e-10 | 657860 | 4.36e-12 |
| $DMR^5_{trans}$ | 0.003% | 0.5% | 181.2 | 0.036 | 48414 | 0.017321 |

**Fig. 4 LDSC results for schizophrenia heritability.** (Top) LDSC analysis outputs for each category of gDMRs are explained here. Results showed that most of the enrichment for schizophrenia heritability in our gDMR sites was in *cis*. (Bottom) Visual description of LDSC: we performed LDSC analysis on the GWAS-identified loci as a background, and two sets of gDMRs, one with a cutoff of *t* > 3.5 and one with a more stringent cutoff of *t* > 5. We further divided these gDMR sets into *cis* gDMRs—those within the GWAS loci—and *trans* gDMRs—those outside the GWAS loci. *P*-values presented here are not adjusted for multiple testing, though multiple testing correction was used to determine significance in these analyses.

sorted with the NeuN antibody and subjected to WGBS[46]. We first performed full genome meQTL analysis on CpH sites, and found a robust presence of CpH-meQTLs in the DLPFC, with 25,584,299 SNP-CpH pairs representing 5,805,754 SNPs, 468,914 of which were not significant meQTLs for CpG sites (see "Data availability" section). These CpH-associated SNPs further had CpG sites nearby, including in the testing window, suggesting potentially independent or complementary effects of CpH and CpG genetic associations. Unlike widespread CpG associations to genotype, there were far fewer unique CpH sites associated with genotype—only 976,094 CpHs associated with genotypes, corresponding to just 1.5% of tested sites. Generally, genetic control on CpH methylation appeared to have a narrower effect than on CpG methylation, with each SNP associating with a mean of 4 CpH sites over a mean width of 12,570 bp. The effect sizes of genotypes on methylation levels were much larger than they were for CpGs, with a mean of 27% change in methylation level per allele, and more than half (57%) of these CpH sites were inside genes. The landscape of CpH-meQTLs in the hippocampus was similar to DLPFC, identifying 25,043,471 SNP-CpH pairs, representing 5,853,364 SNPs and 781,490 CpHs (see "Data availability" section). A large majority (90%)—but not all—of these SNPs and 63% of these CpH sites were also meQTLs in the DLPFC. Similarly, CpH-meQTLs had much larger effect sizes (mean = 29%) and most represented CpH sites (58%) were inside genes. These effects in each brain region presumably represent neuronal-specific genetic regulation of DNAm levels.

We also performed more focused CpH-meQTL analyses on the PGC SNPs described above and found 1444 significant *cis*-meQTLs and 48 *trans*-meQTLs in the DLPFC (Supplementary Dataset 6). Again, a majority of PGC SNPs were represented (141/152). Some of these CpH sites were near CpG-meQTLs, but many were not (mean distance = 120 kb, median = 2798 bp), suggesting potential independent effects of genotype on different sequence contexts of DNAm. Like with CpGs associated with genotype, CpHs in PGC-meQTLs were also enriched for transcriptional and weak transcriptional chromatin states over full genome CpH-meQTLs, and depleted for repressor polycomb and quiescent states

(Fig. 2c)[33]. Most CpHs were inside genes that were subsequently enriched for neuronal GO terms related to neurons, synapses, and channels, further validating the neuronal contribution of CpH DNAm levels. We similarly observed much larger effect sizes of risk alleles in CpHs compared to CpGs in line with genome-wide analyses above, with a mean of 27% compared to 2%, respectively. In the hippocampus, we found 1588 *cis*-CpH-meQTLs and 92 *trans*-CpH-meQTLs (Supplementary Dataset 7), representing 148/153 tested SNPs. Similar to all previous analyses we see that these sites are mostly inside genes and have much larger effect sizes than CpGs. The genes represented by these CpHs are enriched for GO terms related to neuronal anatomy, synapses, and IP3. Again, distance to the nearest CpG-meQTL is highly variable, ranging from 1 to 4,217,747 bp (mean = 24,374, median = 2,761). Results were overall similar between both brain regions, and 1219 CpH-meQTLs were in common between both regions, though again, there were unique associations across regions. Overall, all PGC index SNPs were meQTLs, with most associating with both CpG and CpH sites, but a small percentage only associating with one cytosine context.

**Age associations to DNAm levels**. While there was extensive evidence of meQTLs in our WGBS data, there were a subset of CpG sites that showed high percentages of variance explained by age (Fig. 1f). We therefore more formally modeled methylation over age in both brain regions, as DNA methylation has been shown to globally accumulate with age[47,48]. We found an extensive association with age; nearly 1 million CpGs (DLPFC = 765,861, HIPPO = 972,047) associated with age at FDR < 0.01, and ~2 million CpG sites in each region (at FDR < 0.05, see "Data availability" section). The majority of these sites were age-associated in both regions, with a sizable fraction of sites showing some regional specificity (700,000 sites in the DLPFC and 800,000 sites in the hippocampus). The majority (94%) of sites increase in methylation with age, with half of the sites in promoter regions, and a quarter in CpG islands or shores. Only 9% of genes represented by these differentially methylated

promoters had a significant correlation to gene expression levels in these samples, perhaps resulting from DNAm and RNA-seq data being derived from different tissue dissections, and thus having the different cellular composition (see "Methods" section). In contrast, there was very little CpH association with age, with only 5136 and 445 significant sites in hippocampus and DLPFC respectively (at FDR < 0.05). These results suggest that CpH methylation may be more stable across adulthood and aging after establishment in postnatal life.

Given the large extent of meQTL- and age-associated sites, we asked whether any CpG sites showed dynamic meQTL effects across the adult lifespan. Despite age being associated with methylation at many sites throughout the genome, we found there were practically no changes in meQTL effects across the adult lifespan (i.e., statistical interaction between age and genotype), and, if anything, sites that were differentially methylated by age were depleted ($P < 2.2e-16$) for being associated with local genetic variation (i.e., being meQTLs).

**Minimal illness-state associated differential methylation levels.** We lastly modeled methylation differences between patients with schizophrenia ($n_{DLPFC} = 70$, $n_{HIPPO} = 77$) and neurotypical controls ($n_{DLPFC} = 95$, $n_{HIPPO} = 102$). These associations are typically more subtle—fewer sites with smaller effect sizes—than age or genotype effects in microarray data[10] and more likely to represent cohort- or data set-specific findings[49]. In these WGBS data, we found very few FDR-significant CpG sites—none in DLPFC and 70 in the hippocampus. This is perhaps not surprising based on previous studies and the high multiple testing burden—almost two orders of magnitude more than microarray platforms—and smaller sample sizes due to the expense and computational intensity of WGBS. Re-analysis of our previous DLPFC Illumina 450k microarray data limited to the 164 donors with WGBS data identified an order of magnitude fewer sites than in the full cohort (184 CpGs versus 2104 CpGs at the same Bonferroni-corrected $P < 0.05$), and further, only 4 sites in this re-analysis remained significant enforcing Bonferroni correction using the number of WGBS-tested (rather than microarray-tested) sites, emphasizing the importance of both of these factors in discovery. We found a similar lack of case–control signal at CpH sites, with no significant hits in DLPFC and 1293 in hippocampus, with most (70%) of the hippocampal hits being in or nearby genes. These results suggest that despite major environmental associations of chronic schizophrenia, including smoking, drug treatment, general health deprivations, and chronic psychological stress, the effects observable on the methylome in bulk tissue are remarkably subtle, particularly in the contexts of much stronger genotype- and age-associated effects.

**Discussion**

Here we present the most comprehensive whole-genome bisulfite sequencing (WGBS) study—particularly in the human brain—to date, to better understand technical and biological factors that contribute to genome-wide DNA methylation levels at both CpG and CpH sites. We first demonstrated, at a single base-pair resolution, that meQTLs are highly abundant throughout the entire genome at a breadth and scope previously unseen. Not only can common SNPs associate with CpG methylation, but they also uniquely and independently associate with CpH methylation levels in adult neurons. Furthermore, we demonstrated the clinical relevance of these single base resolution meQTL maps to identify the functional significance of loci identified by GWAS in the human brain. Using schizophrenia as an example, we found DNA methylation associations to nearly every genome-wide significant variant that clustered into many local genetic

differentially methylated regions (gDMRs) that explained significant proportions of disease heritability. We have further created a user-friendly meQTL browser so that other researchers may use this resource to better understand their own genomic regions of interest.

Due to the expense and computational intensity, WGBS is challenging for epigenomic studies. With our large-scale study, we were able to identify the effects of technical and potential biological variables on our data. This has been less well characterized than microarray studies, and we found that batch and ancestry cause much variance in the data, and their effects are exacerbated and alleviated, respectively, by the smoothing process. We also found that ENCODE blacklist regions are unreliable in WGBS data, due to the increased difficulty of alignment[30]. Overall, it is clear that genotype and age impact methylation at a large number of CpG and CpH sites, contrasting with schizophrenia disease state which associates very little with DNAm. There were few SCZD-associated CpGs in the hippocampus and none in DLPFC, potentially in contrast to previous work[10]. We note that our previous microarray study had a much larger sample size (191 cases, 240 controls), and identified CpG probes with very small differences in DNAm levels (<0.05), which were likely smaller than the precision of WGBS with average post-QC coverage of 22 reads (even by increasing precision through smoothing).

Previous studies have identified a genomic presence of meQTLs, but not at a single base-pair resolution. Our findings are largely consistent with previous work, in that meQTLs are indeed extensive throughout the genome, and that most of their regulation occurs locally. However, while earlier estimates reported that 15% of CpGs were under genetic control[11], we greatly increased this fraction to 55%. Like Smith et al.[8], we showed that overlap was generally high between the two brain regions we surveyed, though there are differences as well. Studies have also found that functional meQTLs are enriched for active chromatin states[11] and that meQTLs appear to impact alternative splicing[12], further agreeing with our results and supporting the idea that schizophrenia risk associated loci may represent functional meQTLs. With our large sample and high genomic breadth, we are able to expand on all of these earlier findings at an in-depth genomic level.

These results further implicate DNAm as perhaps the most proximal molecular correlate of DNA sequence variation. The most comprehensive eQTL resource constructed in brain tissue, using over 1400 individuals, identified that ~25% of common genetic variants associated with nearby gene expression levels[50] and our meQTL maps here implicated three times as many SNPs (76%) with a much smaller number of donors. Similarly, the recent GTEx v8 eQTL efforts—performed across 838 donors and 17,382 RNA-seq samples across 49 tissues—implicated 43% of tested SNVs with gene expression in at least one tissue.

These meQTLs further refined our understanding of the functional significance of schizophrenia genetic risk loci. By leveraging WGBS data combined with genotype data from the same samples, we identified molecular phenotypes associated with individual risk variants. This process could more generally filter GWAS findings to regions of the genome that could impart functional consequences of these risk variants. We found that regions that are differentially methylated by risk-associated genotypes explained most of the heritability imparted by the genome-wide significant schizophrenia risk loci, despite spanning a much smaller fraction of the genome (1.6 versus 56.5 Mb). We also found that for some of these risk loci which have been previously identified as eQTLs[39], DNA methylation mediates eQTL effects, refining the potential mechanism by which genetic risk variants may affect brain function. We note that the strongest mediation

effects were seen among exons, indicating that differences in methylation may be key to alternative splicing, as has been previously hypothesized[51]. While our data do not show that DNAm mediates expression for the majority of the meQTLs, this must be viewed with some caution. Our brain samples represent a moment in time in the lifespan of any given brain, and the data are from bulk tissue. At different life stages, perhaps in specific cell populations, mediation effects may be more prominent, particularly in the developing brain[28]. The interplay between sequence variation, DNAm, and gene expression will likely be refined across cell type-specific analyses from the same source tissue.

WGBS also gives the unique ability to examine CpH methylation, an often overlooked mark, particularly in the brain. We found that DNAm levels at specific CpH sites were also associated with genetic variation, which presumably reflected neuron-specific genetic regulation of DNAm levels. It is interesting that the genetic control of CpH methylation seems to have a much larger effect size than that on CpGm, particularly given the fact that the fraction of neurons in our homogenate tissue were uniquely driving these associations. This mark is particularly interesting to examine in relation to psychiatric disorders because it is specific to neurons, so we can point to the cell type of interest at these sites. Understanding which CpHs are under the control of risk loci even further refines our understanding of the risk loci's functions because of this. It is also interesting that despite CpHs being abundant throughout the genome, most meQTL-CpHs are inside genes, possibly further pointing to functional significance. Large-scale analyses in sorted neuronal cell populations can further refine these associations, particularly in different subpopulations of neurons (i.e., inhibitory and excitatory)[52].

Overall, we have established a comprehensive landscape of genetic control of genomic methylation in the human brain. Based on previous findings that many meQTLs are stable across tissue types, a large fraction of this meQTL map could apply to other tissues and cell types. It is clear that genotype has a robust role in determining local methylation levels, not only at CpG sites but at CpH sites as well. These findings can further be applied to understand the functional significance of genetic risk loci identified in GWAS.

## Methods

**Study samples**. Postmortem brain specimens were donated through the Offices of the Chief Medical Examiners of the District of Columbia and of the Commonwealth of Virginia, Northern District to the NIMH Brain Tissue Collection at the National Institutes of Health in Bethesda, MD, according to NIH Institutional Review Board guidelines (Protocol #90-M-0142). Audiotaped informed consent was obtained from legal next-of-kin on every case (as these donations occurred after death, and thus the donors themselves could not consent). Details of the donation process are described previously[53,54]. All adult neurotypical controls were free from psychiatric and/or neurologic diagnoses and substance abuse according to DSM-IV, and had toxicology screening to exclude acute drug and alcohol intoxication/use at the time of death[55].

**WGBS data generation**. Genomic DNA was extracted from 100 mg of the pulverized dorsolateral prefrontal cortex (DLPFC, corresponding to BA46/9) or hippocampus tissue (dissected as previously described[39]) with the phenol-chloroform method. The hippocampus was dissected from the anterior tip posteriorly through to the midbody of the hippocampus at the level of the lateral geniculate nucleus, which included the hippocampus proper (i.e., Ammons Horn and CA1–3) plus the subicular complex. The DLPFC was dissected in a plane perpendicular to the pial surface in area 46 of the cortex to capture from the pial surface to the gray matter–WM junction.

DNA was subjected to bisulfite conversion followed by sequencing library preparation using the TruSeq DNA methylation kit from Illumina. Lambda DNA was spiked prior to bisulfite conversion to assess its rate, and we used 20% PhiX to better calibrate Illumina base calling on these lower complexity libraries. The resulting libraries were pooled and sequenced on an Illumina HiSeq X Ten sequencer with paired-end 150 bp reads (2×150bp), targeting 90 Gb per sample. This corresponds to 30× coverage of the human genome as extra reads were generated to account for the addition of PhiX.

**Data processing**. The raw WGBS data was processed using FastQC to control for quality of reads, Trim Galore to trim reads and remove adapter content[56], Arioc for alignment to the GRCh38.p12 genome (obtained from https://ftp.ncbi.nlm.nih.gov/genomes/all/GCA/000/001/405/GCA_000001405.27_GRCh38.p12/GCA_000001405.27_GRCh38.p12_assembly_structure/Primary_Assembly/assembled_chromosomes/)[57], duplicate alignments were removed with SAMBLASTER[58], and filtered with samtools[59] (v1.9) to exclude all but primary alignments with a MAPQ ≥ 5. We used the Bismark methylation extractor to extract methylation data from aligned, filtered reads[60]. We then used the bsseq R/Bioconductor package (v1.18) to process and combine the DNA methylation proportions across the samples for all further manipulation and analysis[31]. After initial data metrics were calculated, the methylation data for each sample was locally smoothed using BSmooth with default parameters for downstream analyses. CpG results were filtered to those, not in blacklist regions (DLPFC $N = 26,155,085$, Hippocampus $N = 26,301,249$), and those which had coverage ≥ 3. CpHs were filtered to sites that had >3 coverage and non-zero methylation in at least half the samples. Due to an unidentifiable primary source of variance, 11 samples in the DLPFC were dropped before analysis. We also extracted DNA sequence variants from 740 common exonic/coding sites for comparisons to DNA genotyping data to confirm sample identities, as implemented in our SPEAQeasy RNA-seq software[61].

**DNA SNP genotyping**. Genotype data on the 183 unique donors under study were processed and imputed with additional donors across the full LIBD postmortem genotype data[18]. Genetic imputation was performed on high-quality observed genotypes (removing low quality and rare variants) using the prephasing/imputation stepwise approach implemented in IMPUTE2[62] and Shape-IT[63] imputation reference set from the full 1000 Human Genomes Project Phase 3 dataset[64] genome build hg19. Imputation was performed separately by the Illumina microarray platform across the entire brain collection, and study-specific samples were extracted from the imputed genotypes. This study contained three imputation batches, corresponding to the Illumina h650 ($N = 43$), 1 M Duo ($N = 139$), and Omni5 ($N = 2$) platforms. Imputed genotypes for the entire cohort were merged across imputation runs/batches in the Oxford file format as dosages, then converted to plink file format as "hard call" genotypes (treating variants with posterior probabilities < 0.9 as missing). After filtering to just the donors in this study, we retained common variants (MAF > 5%, relative to this sample) that were present in the majority of samples (missingness < 10%) and that were in Hardy–Weinberg equilibrium (at $P > 1 \times 10^{-6}$) using the Plink tool kit version 1.90b3a[65]. Multi-dimensional scaling (MDS) was performed on autosomal LD-independent SNPs (variation inflation factor = 1.25, corresponding to $R^2 < 0.2$) to construct genomic ancestry components on each sample, which can be interpreted as quantitative levels of ethnicity—the first component separated the European and African American samples, for inclusion as potential confounders in the differential methylation analyses described below. We also extracted 740 observed and imputed DNA-genotyped SNPs, and successfully confirmed sample identities against these same variants extracted from the WGBS data.

**Assessment of technical and biological variation**. Principal component analyses (PCA) were performed on the 1e6 most variable autosomal CpG sites using the prcomp() function in R. We calculated the percentage of variance explained by biological and technical variables using the anova() and lm() functions in R.

**meQTL analysis**. We used R package Matrix eQTL[66] (v2.3) in all meQTL analyses. For full genome analysis, we set the maximum *cis* SNP to "gene" distance to 20 kb. We approximated the $P$-value equivalent to FDR = 0.01 and used this as the $P$-value cutoff. We used only SNPs which were common (with minor allele frequencies, MAF > 5%) across the donors in each data set separately that were in Hardy–Weinberg equilibrium (at $P > 1e−6$) with high non-missingness (>90% present), leading to analysis of 7,897,043 SNPs in the DLPFC and 7,865,986 SNPs in the hippocampus. The model adjusted for 28 covariates, which were the top 28 principal components of the methylation data. For PGC SNP analyses, we set the *cis* distance to 250 kb, and considered everything else in trans. We set the $P$-value cutoff to 1 so that we had statistics for every SNP–CpG pair in this analysis. meQTL interaction with age, neuronal composition, and MDS1 was assessed using the modelLINEAR_CROSS parameter. meQTLs were then organized into gDMRs by using the bumphunter R/Bioconductor package (v1.30)[67] function regionFinder, to create clusters of adjacent meQTLs which all had an association statistic of $t ≥ 5$. These were filtered to gDMRs containing at least two adjacent CpGs. We used the cleaningY() function from the jaffelab package[68] version 0.99.20 to regress out adjustment covariates to visualize the DNAm levels in subsequent plots. In order to assess replicability and consistency between meQTL analyses and different sample subgroups, we calculated $\pi_1$ statistics, which estimates the fraction of alternative hypotheses. To do this, we take all significant meQTLs from the main analysis, and then calculate the statistics for these SNP–CpG pairs in the data set we wish to compare. We then take all the $P$-values—regardless of their significance—from the comparison group, and use R package $q$-value[69] (v2.20) to calculate $\pi_0$ (which is the proportion of true null hypotheses), which is then subtracted from 1 to calculate $\pi_1$.

**Heritability analysis**. We estimated the SNP-heritability of DNAm for each CpG site using the GCTA software[35]. We removed seven samples of DLPFC and eight samples of HIPPO so that all pairs of retained samples (DLPFC 158, HIPPO 171) had a genetic relatedness less than 0.025 and were included for heritability estimation. The genetic relationship matrix (GRM) was calculated using SNPs around each CpG site at three different window sizes (40 kb, 200 kb, and 1 Mb). We included the same set of covariates as we used in meQTL analysis in heritability estimation.

**Functional significance analysis**. We annotated our data to nearby genes relative to Gencode v. 29 on hg38. We performed Gene Ontology and gene set enrichment using clusterProfiler (v3.12)[70] with a P-value cutoff of 0.01 and q-value cutoff of 0.05.

**Stratified linkage disequilibrium score regression**. We performed stratified LD-score regression (LDSC) as described by Finucane et al[43], as implemented by Rizzardi, Hickey et al.[71] for defined DMRs using summary statistics from recent GWAS[72]. More detailed methods are provided in Price, Collado-Torres, et al.[46], including code: https://github.com/LieberInstitute/brain-epigenomics/tree/master/LDSC/code. Briefly, we used LDSC (LD SCore) v1.0.0 to estimate the proportion of heritability captured in sets of gDMRs for each GWAS phenotype, as well as central nervous system annotations included in the LDSC package (referred to as CNS (LDSC)), and regions annotated as putatively regulatory in the human brain using chromHMM (i.e., the union of regions annotated as "Bivalent Enhancer," "Bivalent/Poised TSS," "Genic enhancers," "Flanking Active TSS," "Active TSS," "Strong transcription," and "Enhancers." After converting the GWAS summary statistics into the .sumstats format using munge_sumstats.py, we filtered to only HapMap 3 SNPs (downloaded from https://data.broadinstitute.org/alkesgroup/LDSCORE/w_hm3.snplist.bz2) as described in the Partitioned Heritability LDSC tutorial. The partitioned heritability for each gDMR-GWAS combination by adding each feature individually to the "baseline model" including 53 baseline annotations described in Finucane et al.[43].

**DNA methylation mediation of expression**. To assess mediation of gene expression, we identified SNPs which were both eQTLs[39] and meQTLs, and which had some correlation (cor > 0.3) between gene expression and methylation levels. For every CpG-gene pair generated by this, we modeled the effect of additive genotype on expression (Expression ~ SNP), then added in the effect of methylation (Expression ~ SNP + DNAm), and examined the difference in the effect size/coefficient for genotype between the two models. To determine the extent of mediation, we calculated the ratio of SNP coefficient in the second model to the SNP coefficient in the first. When the proportion was <75% (>25% reduction), we considered this evidence of mediation. The same analysis was done for exon and junction expression data and their significant eQTLs.

**Age and diagnosis differential methylation modeling**. Differential methylation analyses for both diagnosis and age were performed using linear regression modeling, accounting for sex, estimated neuronal fraction, batch, and the top three MDS components from genotype data. The regression analyses above were formed using limma (v3.30)[73,74] which employed empirical Bayes and returned moderated T-statistics, which were used to calculate P-values and estimate the false discovery rate (FDR, via Benjamini–Hochberg approach[75]).

**General statistical reporting**. Sample sizes were 165 samples in DLPFC analysis, 179 samples in hippocampus analysis, and 344 samples in combined analyses (161 DLPFC and hippocampus matched pairs from the same donors). All box plots shown in the main and supplementary figures display the median as the center, IQR (25th–75th percentile) as the box range, and 1.5 times the IQR as the whiskers. All reported P-values are two-sided, and multiple testing correction method is noted in the text. Distributions of the residuals of our many linear models were assumed to be normally distributed across all sites and models, but this was not formally tested. All points were used in all analyses: for example, outliers were not removed.

**Reporting summary**. Further information on research design is available in the Nature Research Reporting Summary linked to this article.

## Data availability
We have created a user-friendly and fast meQTL browser that allows searching by SNPs or cytosines by genomic regions (chr.start-end) at https://eqtl.brainseq.org/WGBS_meQTL/. Raw and processed nucleic acid sequencing data generated to support the findings of this study are available via the PsychENCODE Knowledge Portal (https://psychencode.synapse.org/). The PsychENCODE Knowledge Portal is a platform for accessing data, analyses, and tools generated through grants funded by the National Institute of Mental Health (NIMH) PsychENCODE program. Data is available for general research use according to the following requirements for data access and data attribution: (https://psychencode.synapse.org/DataAccess). For access to content described in this

manuscript see: https://doi.org/10.7303/syn25992404. Full results data sets can be found in the same repository or in the Supplement's "Description of Additional Supplementary Files". Due to containing identifiable information, genotype data are available through controlled access via the corresponding authors following successful access to dbGaP data set phs000979.

## Code availability
Analysis code that accompanies this paper is provided on GitHub (https://github.com/LieberInstitute/wgbs_meqtl_analysis) and available at Zenodo (https://doi.org/10.5281/zenodo.5113698)[76].

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

## Acknowledgements

We would like to express their gratitude to our colleagues whose tireless efforts have led to the donation of postmortem tissue to advance these studies: the Office of the Chief Medical Examiner of the District of Columbia; the Office of the Chief Medical Examiner for Northern Virginia, Fairfax Virginia; and the Office of the Chief Medical Examiner of the State of Maryland, Baltimore, Maryland. We would also like to acknowledge Llewellyn B. Bigelow, MD, for his diagnostic expertise. This project was supported by The Lieber Institute for Brain Development and by NIH grants R01MH112751 and T32GM781437. Finally, we are indebted to the generosity of the families of the decedents, who donated the brain tissue used in these studies.

## Author contributions

K.A.P.M. and A.E.J. conceptualized the project and methodology, investigated and analyzed the data, and wrote the paper. S.H. aided in analysis. W.S.U., L.C.-T., A.S.S., R.W. and N.J.E. provided software support. W.S.U. created the online meQTL browser. N.J.E., A.J.P., R.T., T.M.H. and R.W. curated data. J.E.K., R.T. and T.M.H. provided resources. S.H., A.J.P., L.C.-T., S.A.S., A.E.J. and D.R.W. reviewed and edited the paper. D.R.W. provided supervision, and A.E.J. was the principal investigator and oversaw experimental design and analysis. All authors read and approved the final manuscript.

## Competing interests

Andrew E. Jaffe is now employed by a for-profit biotechnology start-up company (with company name pending), which is unrelated to the contents of this manuscript. The remaining authors declare no competing interests.

**Additional information**

