## [Peer Review File · Nature Communications]

Reviewers' Comments:

Reviewer #2:

Remarks to the Author:

The authors have addressed all of my questions. The only additional suggestion is that the authors mention in the manuscript the meQTL analysis stratified by diagnosis, which they carried out in response to my point #2. It is a useful control to include.

Reviewer #4:

Remarks to the Author:

In general, the Authors responded to previous review adequately.

From my perspective, the primary strength of the paper is that it provides a unique resource to the field, with brains from 181 donors whole genome bisulfite sequenced, including 161 subjects with both prefrontal cortex and hippocampal tissue assayed.

The result section is ok, albeit results as presented as heavily dependent on the way the analyses is conducted.

The one issue I would like to raise that the paper is not very transparent about how many disease cases (Schizophrenia) and how many controls were included, or at least I have a hard time finding that information.

This is important as in their results chapter, 'Minimal illness-state associated differential methylation levels', they essentially report a negative finding comparing cases versus controls and explain this by genotype and age-associated effects playing a much stronger role.

That may be so, but I feel the Authors have the duty to discuss more in detail how to reconcile this current finding with their Jaffe et al. 2016 paper in Nature Neuroscience with 2,104 CpG altered in their disease cohort back then.

Simply, as the Authors do now, explaining away their 2016 findings as technical phenomenon due to higher burden of multiple testing in WGBS (as opposed to array/bead technique back then) is insufficient.

At least they could track the 2,104 GpGs (From the 2016 paper) in the current study, and correct for age and genotype effects, etc, to check if they can replicate their findings from back then.

Reviewer #2 (Remarks to the Author):

1. The authors have addressed all of my questions. The only additional suggestion is that the authors mention in the manuscript the meQTL analysis stratified by diagnosis, which they carried out in response to my point #2. It is a useful control to include.

We have now added a sentence to our results subsection titled "Local genetic variation has strong effects on CpG DNA methylation levels" describing the results of these analyses. "As an additional sensitivity analysis, we performed meQTL analysis within each diagnostic group separately to assess the extent by which schizophrenia diagnosis influenced our findings. We found extremely high sharing ($\pi_1 > 99\%$) among significant meQTLs discovered in either diagnostic groups (and assessed in the other/non-discovery group), supporting that meQTL effects were globally similar between diagnostic groups."

Reviewer #4 (Remarks to the Author):

In general, the Authors responded to previous review adequately.

From my perspective, the primary strength of the paper is that it provides a unique resource to the field, with brains from 181 donors whole genome bisulfite sequenced, including 161 subjects with both prefrontal cortex and hippocampal tissue assayed.

The result section is ok, albeit results as presented as heavily dependent on the way the analyses is conducted.

We thank the reviewer for their time and effort in reviewing our revised manuscript.

2. The one issue I would like to raise that the paper is not very transparent about how many disease cases (Schizophrenia) and how many controls were included, or at least I have a hard time finding that information. This is important as in their results chapter, 'Minimal illness-state associated differential methylation levels', they essentially report a negative finding comparing cases versus controls and explain this by genotype and age-associated effects playing a much stronger role.

We have now added this information at the start of the Results subsection titled: "Minimal illness-state associated differential methylation levels", specifically modifying this sentence: "We lastly modeled methylation differences between patients with schizophrenia ($n_{\text{DLPFC}} = 70$, $n_{\text{HIPPO}} = 77$) and neurotypical controls ($n_{\text{DLPFC}} = 95$, $n_{\text{HIPPO}} = 102$)."

3. That may be so, but I feel the Authors have the duty to discuss more in detail how to reconcile this current finding with their Jaffe et al. 2016 paper in Nature Neuroscience with 2,104 CpG altered in their disease cohort back then. Simply, as the Authors do now, explaining away their 2016 findings as technical phenomenon due to higher burden of multiple testing in WGBS (as opposed to array/bead technique back then) is insufficient. At least they could track the 2,104 GpGs (From the 2016 paper) in the current study, and

correct for age and genotype effects, etc, to check if they can replicate their findings from back then.

There were several key differences between the 2016 microarray-based study and the present WGBS study. First, the prior study had a much larger sample size, due to the lower cost of microarrays vs WGBS: 191 adult patients with schizophrenia (compared to 70 here) versus 240 neurotypical controls (compared to 95 here). This prior study was presumably more highly powered to detect SCZD-control differences, which had small effect sizes - all differences in DNAm levels were < 0.05 in the 2016 paper. To explore whether decreased sample sizes in the present paper contributed to fewer case-control differentially methylated sites, we re-analyzed the Illumina 450k data from the 2016 paper after filtering to the 164 subjects profiled with WGBS (from 165 total samples; one sample was not profiled on the microarray). With this reduced sample size on the Illumina 450k, only 184 CpG probes were differentially methylated between SCZD versus controls (at $p_{\text{bonf}} < 0.05$, analogous to the published results from the full cohort), which was an order of magnitude fewer sites identified. Furthermore, only 541 sites on the Illumina 450k in the full cohort remained significant after controlling for the number of WGBS sites we profiled (27M), and only 4 sites remained when adjusting to this cutoff in the 164-sample analysis. This sensitivity analysis supported our hypothesis that the lack of power is a major reason for lack of replication. Second, all of the DNAm case-control proportion differences on the Illumina 450k had effect sizes less than 0.05. These small effects were likely identified due to the increased power of the larger sample size, and less noisy Illumina 450k measurements (albeit at a much small number of sites across the genome). In our WGBS data, the average post-QC read was 22, and although we gained precision through smoothing, DNAm differences less than 1 read (ie $1/22$, 0.045) would be difficult to detect. We do note that both WGBS and microarray differential methylation models for SCZD diagnosis already adjusted for global ancestry (via MDS components calculated from genotype data) and age, suggesting differences in modeling did not contribute to differences in discovery. We have added several of these caveats in the Discussion and Results sections.

Reviewers' Comments:

Reviewer #4:

Remarks to the Author:

The Authors responded to the remaining issues raised in re-review and I don't have additional comments.